# A Novel Cognition of Decitabine: Insights into Immunomodulation and Antiviral Effects

**DOI:** 10.3390/molecules27061973

**Published:** 2022-03-18

**Authors:** Ji Xiao, Ping Liu, Yiliang Wang, Yexuan Zhu, Qiongzhen Zeng, Xiao Hu, Zhe Ren, Yifei Wang

**Affiliations:** 1Jinan Biomedicine Research and Development Center, Department of Cell Biology, College of Life Science and Technology, Jinan University, Guangzhou 510632, China; jixiao_1992@163.com (J.X.); 18370415798@163.com (P.L.); yiliang_wang@foxmail.com (Y.W.); zhuyexuan0917@163.com (Y.Z.); zqiongz@stu2019.jnu.edu.cn (Q.Z.); huxiao3520@163.com (X.H.); 2Guangdong Province Key Laboratory of Bioengineering Medicine, Jinan University, Guangzhou 510632, China; 3Guangdong Provincial Biotechnology Drug & Engineering Technology Research Center, Jinan University, Guangzhou 510632, China; 4National Engineering Research Center of Genetic Medicine, Jinan University, Guangzhou 510632, China; 5College of Pharmacy, Jinan University, Guangzhou 510632, China; 6Biomedicine Research and Development Center, Jinan University, Guangzhou 510632, China

**Keywords:** DNA methylation, decitabine, antiviral innate immunity, interferon

## Abstract

DNA methylation, as one of the major means of epigenesis change, makes a large difference in the spatial structure of chromatin, transposable element activity and, fundamentally, gene transcription. It has been confirmed that DNA methylation is closely related to innate immune responses. Decitabine, the most efficient available DNA methyltransferase inhibitor, has demonstrated exhilarating immune activation and antiviral effects on multiple viruses, including HIV, HBV, HCV, HPV and EHV1. This review considers the role of decitabine in regulating innate immune responses and antiviral ability. Understanding the complex transcriptional and immune regulation of decitabine could help to identify and validate therapeutic methods to reduce pathogen infection-associated morbidity, especially virus infection-induced morbidity and mortality.

## 1. Background

Epigenetic regulation occurs at multiple levels, including through DNA methylation, histone modification, RNA interference, nucleosome remodelling and modulation of the 3D chromatin structure, and contains almost all molecular mechanisms affecting gene expression in a reversible, transmissible, and adaptive way without altering the genomic DNA sequence [1]. These dynamic epigenetic regulations play a significant role in transcriptional regulation, genomic integrity, cell fate and physiological control of tissue and organ development. DNA methylation is established through the addition of a methyl group from S-adenosyl-methionine to the 5′ position on a cytosine within a cytosine–guanine (CG) dinucleotide. In general, CpG islands are rare in mammalian DNA with a typical expected: observed ratio of 30% or lower. In promoters with CpG islands, this ratio is 60% or higher [2]. Promoter hypermethylation is an epigenetic mechanism of gene regulation known to silence gene expression. Various factors, such as ageing, differentiation, and environmental stress, can alter DNA methylation patterns in mammalian cells [3], including immune-related cells [4]. DNA methylation is deposited and maintained through the concerted activity of three essential DNA methyltransferases, mainly DNMT1, DNMT3A and DNMT3B [5]. Mounting evidence suggests that 5-aza-2′-deoxycytidine (5-AZA-dC, decitabine, DAC) (Figure 1A), the most widely used inhibitor of DNA methyltransferases (DNMTs), induces demethylation of DNA, leading to consecutive reactivation of epigenetically silenced tumour suppressor genes mainly in practice for haematological tumours, and it is being developed for solid tumours. While exploring the antitumour effect of DAC, an array of studies has revealed that in addition to inhibiting cell proliferation, inducing cell apoptosis and regulating tumour immunity, DAC shows a crucial function in the innate immune response [6,7,8,9]. Moreover, as a nucleic acid analogue, DAC has demonstrated potential antiviral activity by upregulating innate antiviral immune responses. In 2006, decitabine (DAC) was approved by the FDA for the treatment of patients with myelodysplastic syndrome (MDS) [10]. Therefore, it is highly valuable to explore its new applications in addition to antitumour functions, especially in antiviral activity.

## 2. DNA Methylation in Antiviral Innate Immune Response

Three primary mechanisms are considered to induce epigenetic changes: DNA methylation, histone modification, and noncoding RNA-associated gene regulation. Herein, DNA methylation is widely related to chromatin modification, given its pivotal role in gene silencing, X-chromosome inactivation, genomic stability and imprinting in mammalian cells, and occurs in cells via the addition of methyl groups by DNA methyltransferases (DNMTs) at position 5 of cytosine residues at CpG-rich promoter regions to silence specific genes such as tumour suppressor genes (Figure 1C). There are approximately 28 million CpG dinucleotides in human genomic DNA, reaching 60–80% methylation in any given cell, but CpG dinucleotide-enriched regions known as CpG islands, mostly located in or near gene promoter sequences, are predominantly hypomethylated [11]. DNA methylation patterns are regulated by the function of a family of DNA methyltransferases -DNMT1, DNMT3A and DNMT3B. Unmethylated DNA is methylated by de novo DNMT3A and DNMT3B, which have ubiquitous and nonselective activity. The maintenance DNA methyltransferase DNMT1 binds the hemimethylated DNA strand and copies with the parental strand CpG methylation pattern to the daughter strand [5]. The immune system is responsible for host immune surveillance, defence and regulation, and primordial functions of living organisms, with the goal of preserving tissue and organismal homeostasis. Infectious diseases, especially viral infections, remain a serious threat to human health, and the host innate immune system exhibits a critical defence against pathogen invasion. Innate immunity is the first line of defence against pathogens and infections, and the innate immune system is made up of tissue barriers (skin and mucosal system, blood–brain barrier, placental barrier, etc.), innate immune cells (phagocytes, natural killer cells, dendritic cells, etc.) and innate immune molecules (complement, cytokines, enzymes, etc.) [12]. Among them, the antiviral innate immune response has a direct and effective inhibitory effect on viral infection and replication. DNA methylation, as a transcriptional regulator of the immune system, makes a great difference in immune system development, differentiation and function, and aberrant DNA methylation is the major epigenetic change leading to oncogenesis [13]. Moreover, DNA methylation is closely linked to the antiviral immune response. Recent research found that Uhrf1 (one of the critical molecules involved in DNA methylation) deficiency in myeloid cells significantly upregulated IFN-β expression, increasing resistance to viral infection. The whole-genome bisulfite sequencing revealed that a single-nucleotide methylation site in the Ifnb promoter region disrupted IRF3 recruitment [14]. Inhibition of DNA methylation can induce an antiviral response through upregulation of the interferon signalling pathway. Multiple molecular mechanisms, such as activating endogenous retroviruses, initiating key factor expression in the interferon signalling pathway, regulating the NF-κB pathway and enhancing regulatory T cell function, are involved in the DNA methylation-mediated antiviral response [2,6,7,9,15,16,17,18]. Viral infections can alter the epigenetic landscape through modulation of DNA methylation profiles. Herpes simplex virus type 1 (HSV-1) capsid protein VP26 was identified to bind to the host factor de novo DNA methyltransferase DNMT3A during infection, and downregulating DNMT3A with siRNA or suppression by the human DNMT inhibitor RG108 can dramatically trigger a decrease in HSV-1 titres [19]. Several possible mechanisms are involved in the effect of DNMT3A on HSV-1 infection, and immune recognition probably participates in this process [6,7,20]. The DNMT inhibitor 5-aza-2′-deoxycytidine was identified to induce a retinoic acid-inducible gene I (RIG-I/DDX58)-related innate immune response, one of most significant anti-dsRNA virus immune signalling pathways, by modulating mitochondrial stress in neuroblastoma [15]. In vitro methylation of the interferon regulatory factor-7 (IRF-7) gene promoter blocks its expression. Regulating the promoter activity of IRF-7 by 5-AzaC (5′-aza-deoxycytidine) was sufficient to elicit basic levels of IRF-7 expression and further stimulate the Type I IFN (interferon) pathway [2]. Toll-like receptors (TLRs) play an important role in the antiviral innate immune response. Upon detailed analysis of the TLR2 promoter CpG island, higher CpG methylation was detected in gingival epithelial cells. When cells were treated with a DNA methyltransferase inhibitor, TLR2 mRNA and downstream innate immune-related cytokine expression were significantly upregulated [21]. DNA methylation for *TLR-2* and *TLR-9* in spontaneous preterm labour (sPTL) was notably reduced compared to term not in labour (TNL) or term in labour (TL). Treatment of THP-1 cells with 5-Aza resulted in remarkable increases in *TLR-2* and *TLR-9* mRNA expression, which was associated with a noteworthy upregulation in the expression of the neutrophil chemokine IL-8 [22]. An array of studies has proposed that DAC treatment and DNMT1 knockdown induce an antiviral response, including activation of interferon-responsive genes via dsRNA-containing endogenous retroviruses in cancers [6,7,13,17]. In conclusion, methylation modifications of genomic DNA and some specific innate immune-related genes in the prompter or body have been shown to be closely associated with antiviral innate immune responses.

## 3. Decitabine and Its Role in the Immune Modulation of Viral Diseases

In 1964, the azanucleosides 5-azacytidine (azacytidine, AZA) and 2′-deoxy-5-azacytidine (5-aza-dC, decitabine, DAC) were first synthesized as classical cytostatic agents [23] (Figure 1A,B). As nucleic acid analogues, detection results of the in vitro stability of decitabine in a neutral aqueous solution indicated considerable chemical stability (half-life time of 7 days at 4 °C, 96 h at 20 °C and 21 h at 37 °C), and even storing the solution at room temperature effectively inhibited DNA methylation. At 37 °C, the half-life times were 7 h for azacytidine and 21 h for decitabine [24]. DNMT inhibitors (DNMTis), such as 5-azacytidine (azacytidine) and decitabine, are the most frequently used epigenetic modulators employed in routine clinical practice for the treatment of malignant diseases. DAC can have direct or indirect effects on gene expression. The effect is direct when DAC incorporation into a gene significantly alters its methylation and expression status; promoter and gene body demethylation are two such examples. The effects are considered indirect when gene expression is altered without the gene itself undergoing any marked change in methylation.

As a cytosine analogue, DAC can be incorporated into DNA and trap DNA methyltransferases (DNMTs), resulting in their proteasomal degradation and global DNA demethylation [25]. As mentioned in the preceding part of the text, DNA methylation is a dynamic epigenetic modification with a prominent role in the immune system [13], indicating that decitabine, one of the most effective inhibitors of DNMTs, plays an important role in the regulation of the interferon signalling pathway. Both DAC and DNMT1 siRNA caused overall hypomethylation, and hypomethylation at the promoters of many histones and hypermethylation at multiple sites genome wide were unique to DAC treatment [8]. DAC has been shown to be a powerful inducer of human endogenous retrovirus, HERV-Fc1 in cells previously not expressing HERV-Fc1, or with a low expression level, and at the same time, it strongly inhibits methylation of DNA [26]. Transient treatment of HCT116 colorectal cancer cells with a low dose of DAC induces an increase in dsRNAs and durable DNA demethylation-independent activation of the det gene enriched for interferon-responsive genes and the MDA5/MAVS/IRF7 pathway [7]. Additionally, DAC-induced transcripts of human endogenous retroviruses (ERVs), which constitute more than 8% of the human genome, can activate interferon signalling-mediated viral defence responses in epithelial ovarian cancer (EOC) [6]. Hung-Yu Lin reported that DAC effectively induced a RIG-I-related innate immune response and apoptotic signalling primarily in SK-N-AS NB (human neuroblastoma cells) cells by hypomethylating the DDX58/RIG-I promoter, elevated mtROS and increased dsRNA [15]. These reports suggest that decitabine is a promising compound for innate immune response regulation. In addition, inflammation affects immunoregulation. Bioinformatics analysis showed upregulated DNMT1 expression and suggested upregulated NF-κB signalling pathway-related genes in patients with sepsis. Degrading intracellular DNMT protein levels by decitabine improved the inflammatory response and survival in mice with severe sepsis induced by caecal ligation and puncture (CLP) [18]. GO (gene ontology) analysis of the genes demonstrated that IKK/NF-κB cascade-related genes such as Bst2, Rnf31, Zc3hav1 and Ubd were dramatically upregulated upon inhibition of DNA methylation with 1 μM DAC on colon tumour organoids [9]. Low-dose decitabine treatment enhanced IκBα degradation and induced NF-κB activation in CD4 T cells from patients with a response to decitabine-primed chemotherapy rather than those without a response [27]. DAC also regulated the inflammatory response by the significant upregulation of p-IKKα/β, p-IκBα, p-p65, p-p38 and p-ERK in lipoteichoic acid (LTA)-stimulated human odontoblast-like cells (hOBs) [28]. A recent study showed that, in B cell lymphomas, decitabine repressed B cell-specific gene transcription and activated NF-κB signalling; during osteoclastogenesis, decitabine conversely inhibited the activity of NF-κB, AP-1 and extracellular signal-regulated kinase (ERK) but not the PI3K/Akt pathway [29]. Taken together, decitabine showed a shifting function on the NF-κB pathway, mainly regulating the inflammatory response, but showed a concentrated character on the interferon response pathway, which makes decitabine an ideal drug candidate for interferon-related diseases such as pathogen infection.

## 4. Antiviral Effects of Decitabine

### 4.1. HIV

Human immunodeficiency virus (HIV)-induced acquired immunodeficiency syndrome (AIDS) has persevered for more than four decades, but ongoing global challenges remain. Epigenetic modification plays a vital role in the life cycle of HIV [30]. In 1990, Bouchard, J. et al. found that two 5-azacytidine derivatives, 5-azacytidine (AZA) and 5-deoxyazacytidine (DAC), could effectively inhibit the replication of HIV-1 in the human T-lymphocyte cell line CEM cells without cytotoxicity. The potential mechanism may involve demethylation of viral DNA. The methylation of viral DNA modulates the expression of HIV proviruses and leads to the instability of HIV provirus DNA [31]. Studies have shown that DAC, a reducer of AZA, is more effective in inhibiting HIV, mainly through the enhancement of HIV-1 mutations [32]. It has been reported that fatal mutations of HIV-1 mediated by decitabine are related to base-G and C mutations, and the G-to-C mutation mediated by decitabine is an effective antiviral mechanism to inhibit HIV-1 infection [33]. Through the synthesis of new resveratrol derivatives, it was found that it not only had anti-HIV-1 activity but also had a strong synergistic effect of resveratrol combined with decitabine. The mechanism of synergistic reduction of HIV-1 infection by resveratrol and decitabine may be related to their ability to enhance cytotoxicity [34]. Clouser, C.L. et al. used drug retargeting to identify clinically approved drugs with anti-HIV activity. The results showed that the combination of two clinically approved drugs, decitabine and gemcitabine (4-Amino-1-[(2R,4R,5R)-3,3-difluoro-4-hydroxy-5-(hydroxymethyl)oxolan-2-yl]pyrimidin-2-one, a difluorine nucleoside analog), synergistically reduced HIV infectivity at concentrations significantly lower than those used for cancer treatment. A potential drug target could be the mutation rate of HIV [35]. In addition, by using the murine leukaemia virus (MuLV)-based LP-BM5/murine AIDS (MAIDS) mouse model (LP-BM5/MAIDS), the authors found that the combination of decitabine and gemcitabine exhibited strong antiretroviral HIV-1 activity in vivo and in vitro. The main involved mechanisms included the following: (1) Decitabine is integrated into HIV-1 DNA during reverse transcription to form atypical base pairs, thereby increasing the mutation frequency of G-to-C. (2) The anti-HIV-1 activity of gemcitabine is attributed to its inhibition of ribonucleotide reductase, which in turn alters the deoxyribonucleoside triphosphate (dNTP) pool and ultimately enhances the activity of decitabine or increases the frequency of HIV-1 mutations [36]. In the same year, the authors first demonstrated that the divalerate prodrugs of both decitabine and gemcitabine were permeable and stable, and both showed strong anti-HIV-1 activity at noncytotoxic concentrations. Compared to the parent compound, the prodrug increases intestinal permeability, which is usually associated with increased bioavailability and pharmacokinetic properties [37]. In addition, a sequential combination of demethylation agents (DACs) and histone deacetylase inhibitors (HDACIs) or TNFα has been reported to reactivate latent HIV [38]. The synergistic effect of this combination therapy may be explained by the synergistic targeting of epigenetic mechanisms between the two by the inhibition of DNA methyltransferase (DNMT) activity or by the unique pharmacological properties inherent to HDACIs [39]. The functions of decitabine in the process of HIV infection are multifarious. Further exploration of the molecular regulation of decitabine will contribute to the development of antiviral drugs.

### 4.2. Hepatitis Viruses

Hepatitis virus is the pathogen of viral hepatitis. Human hepatitis viruses can be classified as types A, B, C, D, E and G. As a kind of infectious pathogen, hepatitis viruses subject human beings to both physical and psychological torment, and patients often suffer unspeakably. To date, there is no specific clinical drug for hepatitis virus. Therefore, it is important to develop specific new drugs against hepatitis virus. Studies have shown that hepatitis B virus (HBV) can downregulate apolipoprotein A1 (ApoA1) through epigenetic silencing of ApoA1 gene expression by inducing DNA CpG island hypermethylation. Further studies showed that upregulation of DNMT1, DNMT3A, and DNMT3B resulted in hypermethylation of the ApoA1 promoter. Since cholesterol levels are necessary for HBV infection and escape from the host cell membrane, ApoA1 may inhibit HBV expression by inhibiting cellular cholesterol levels, which provides a sufficient theoretical basis for the clinical diagnosis and treatment of chronic hepatitis B (CHB) [40]. Chen, C. et al. identified DNMT1 and DNMT3B as host factors involved in HCV reproduction by using lentivirus-mediated shRNA interference technology. In addition, the DNMT inhibitors AZA and DAC significantly inhibited HCV cell culture (HCVcc) infection, viral RNA replication, and protein expression. These results suggest that DNMTs may be an effective target for HCV infection [41]. Studies have shown that the HCV core protein inhibits E6-associated protein (E6AP) expression through DNA methylation, protects itself from ubiquitin–proteasome degradation and stimulates viral proliferation. However, when E6AP was ectopically expressed, the DNA methyltransferase (DNMT) inhibitor DAC or DNMT1 was knocked out, the E6AP level recovered, and the effect of the HCV core protein on E6AP almost completely disappeared, which provided a potential target for the development of anti-HCV drugs [42]. All the results listed above indicate that decitabine has the potential to be developed into an anti-hepatitis virus therapy.

### 4.3. Other Viruses

In 1995, it was first reported that in the presence of DAC, the production of infectious *Autographa californica* nuclear polyhedrosis virus (AcMNPV) was only slightly affected, while the synthesis of late proteins (polyhedrin and pl0) was abolished, similar to the methyltransferase inhibitor 3-deazaadenosine (3DA-Ado) [43]. It has also been reported that when applied a few hours prior to virus inoculation, DNA methyltransferase inhibitors (AZA, azacytidine and DAC, decitabine) improved baculovirus-mediated gene expression by fourfold or more in all four mammalian cell lines (CHO, CNE, HEK293, and HepG2 cells) [44]. In addition, it has been reported that human parvovirus B19 (B19V) formed chromatin-like structures after cotransfection of B19V-infected clones and pHelper plasmids into HEK293T cells for 12 h. More importantly, DAC treatment reduced the formation of chromatin-like structures and the replication of the B19V genome, suggesting that DNA methylation status may be responsible for the reduced replication of the viral genome and altered RNA processing [45]. Thieulent, C. et al. screened 2891 compounds for resistance to Equid herpesvirus-1 (EHV-1) using the impedance method and found that 22 compounds were effective against EHV-1 in vitro. Among them, valganciclovir, ganciclovir, decitabine, aphidicolin, idoxuridine and pritelivir (BAY57-1293) were identified as the most effective compounds. Valganciclovir and decitabine have synergistic effects. Based on the results, the authors hypothesize that the mechanism of decitabine against EHV-1 is the integration of decitabine into EHV-1 DNA and/or blockage of viral polymerase, thereby inhibiting viral growth [46]. HPV16 is the most likely carcinogenic genotype of high-risk human papillomavirus. Morel, A. et al. found that treatment with the demethylating agent DAC in HPV 16-positive Ca Ski and SiHa cells could inhibit the expression of viral oncoprotein E6 at the mRNA and protein levels and simultaneously upregulate the expression level of miR-375, a tumour suppressor miRNA known to target HPV 16 E6/E7 mRNA. The mechanism of action may involve the demethylation of the miR-375 promoter [47]. Greggs, W.M. et al. demonstrated the activity of four FDA-approved anti-HIV-1 active drugs, tenofovir, raltegravir, decitabine, and gemcitabine, against feline leukaemia virus (FeLV) at nontoxic concentrations. Both HIV-1 and FeLV are retroviruses, and the mechanism of action of gemcitabine, tenofovir and raltegravir against FeLV may be similar to that of anti-HIV-1. However, the fact that FeLV reverse transcriptase (RT) has a higher fidelity than HIV-1 RT indicates that decitabine may act through a different mechanism than anti-HIV-1 [48]. In summary, DAC has antiviral effects on both RNA and DNA types of viruses (Table 1). According to current research, DAC regulates only DNA methylation, and it has not been reported that DAC can affect RNA modification. This indicates that DAC inhibits viruses mainly by regulating host DNA methylation. In addition, the nucleic acids of pathogenic microorganisms have a low methylation level, which further suggests that DAC mainly regulates the immune response by inhibiting the methylation of host DNA. With increasing reports on antiviral research on decitabine, the immune regulation mechanism and antiviral effect of decitabine will advance its antiviral drug development process.

## 5. Conclusions and Future Prospects

As one of the DNMTs, DAC interferes with the epigenetic control of gene expression in cells by impeding DNMTs. DAC can reactivate epigenetically silenced genes and has a role in cancer chemotherapy. However, DAC is also a nucleic acid analogue that shares analogous functions with other nucleic acid antiviral drugs, such as acyclovir (ACV) [49] and ganciclovir (GCV) [50], and much more than this, DAC is able to regulate the antiviral innate immune response in various tumour cells. These results have endowed DAC with particular and promising functions in antiviral therapy regimens. In addition, DAC was approved for the treatment of myelodysplastic syndrome subtypes by the FDA in 2006 and Europe in 2009 and progressively spread to different countries worldwide [10], indicating its obvious advantages in medicinal properties. To date, DAC has been reported to have antiviral effects on HIV, hepatitis virus, EHV1, B19V, HPV16 and FeLV. These studies suggest that decitabine may share a similar function in other types of viruses, and because it is a listed drug, decitabine may be a potential drug for antiviral therapy.

In summary, DAC possesses a spectrum of antiviral activity. However, it is difficult to achieve stable pharmacokinetics with decitabine because of their rapid deamination by cytidine deaminase in vivo and spontaneous hydrolytic cleavage. With the improved understanding of the DAC mechanism of action, researchers have discovered that even nanomolar doses could achieve effective inhibition of DNA methylation while also improving tolerability [51]. Decitabine has demonstrated rapid deamination by cytidine deaminase in vivo and spontaneous hydrolytic cleavage. Developing more stable derivatives of decitabine is a demanding prompt solution. 5′-O-trialkylsilylated DACs-OR-2003 and OR-2100 were confirmed to completely deplete DNA methyltransferase 1 and induce both gene-specific and genome-wide demethylation and were comparable to that of DAC, with fewer adverse effects in vivo [52]. Guadecitabine (SGI-110), an investigational drug for the treatment of myelodysplastic syndrome and acute myeloid leukaemia, is a second-generation DNA methylation inhibitor that was designed to overcome the instability of DAC, with the potential to improve pharmacodynamics, clinical efficacy, and safety [53]. At the same time, as an antitumour drug, DAC’s main role is to inhibit cell growth and induce cell apoptosis. Therefore, high-dose and high-frequency administration has a certain toxicity and side effects. Therefore, reducing the side effects of DAC is an important development direction, and the role of guadecitabine in the field of antiviral therapy is worthy of further exploration. The effectiveness of decitabine therapy is also influenced by the relative transport capacities of the target tissue, and four different classes of proteins participate in the transportation process of nucleosides across membranes in human cells [54]. There was also a statistically significant correlation between the expression level of the equilibrative transporter ENT-1 and the sensitivity of mononuclear cells cultured in vitro from acute myelocytic leukaemia (AML) patients [55]. Therefore, it is a new direction to explore the relationship between host cell nucleotide transporter proteins such as ENT-1 and viral infection. Correspondingly, viruses have also evolved various mechanisms to evade host immunity to ensure efficient viral replication and persistence. Several viruses, such as Ebola virus (EBV), HBV, HPV and Kaposi’s sarcoma-associated herpesvirus (KSHV), can modulate host DNA methyltransferases for epigenetic dysregulation of immune-related gene expression in host cells [56]. Hypomethylation of CpG islands in the interferon regulatory factor 5 (IRF-5) promoter was observed in EBV type III latent infected Burkitt’s lymphoma and gastric carcinoma cell lines to restrain IFR5 expression [57]. Further detailed explorations are required for a more thorough understanding of the molecular mechanism of decitabine immunoregulation and feasible treatments for virus infection.

## Figures and Tables

**Figure 1 molecules-27-01973-f001:**
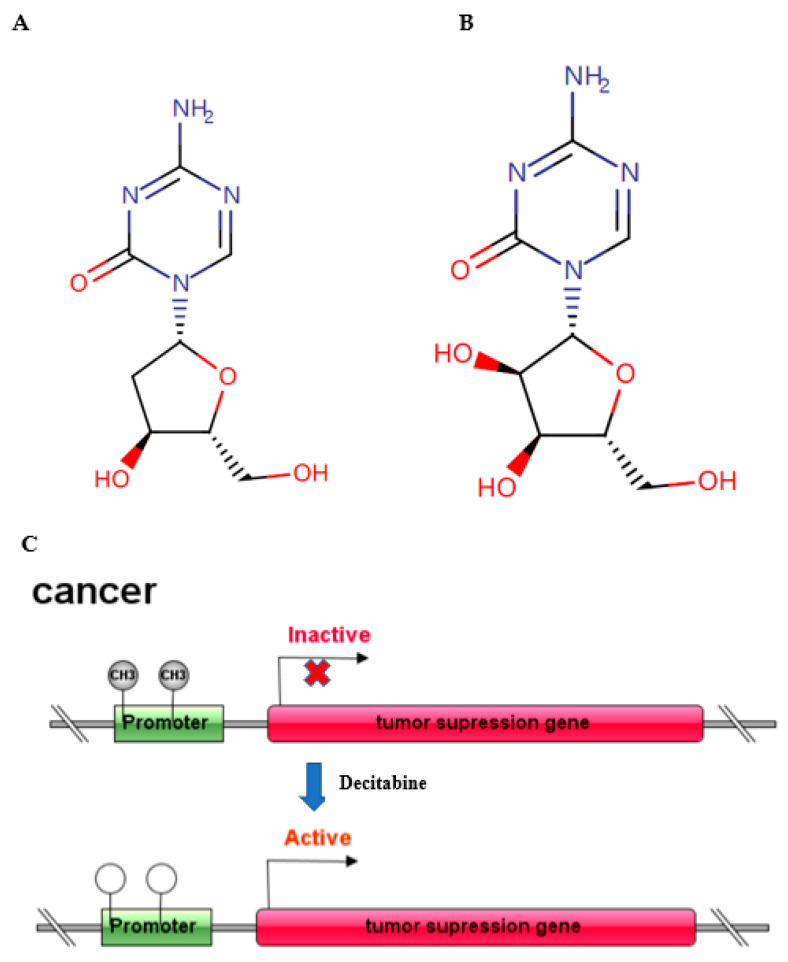
Molecular structure of decitabine (**A**) and 5-azacytidine; (**B**) schematic diagram of antitumor mechanism (**C**).

**Table 1 molecules-27-01973-t001:** Decitabine antiviral effects and potential mechanism.

Virus	Nucleic Acid Type	Potential Mechanism	References
HIV	RNA	Regulated the methylation of viral DNA, leading to the instability of HIV provirus DNA	[31,32,33,34,35,36,37,39]
B19 V	DNA	Regulated the formation and modification of chromatin-like structure	[45]
HBV	DNA	Upregulated anti-inflammatory protein ApoA1	[40]
EHV1	DNA	Integrated into the EHV-1 DNA and/or jams the viral polymerase	[46]
HPV16	DNA	Upregulated miR-375 level and represses HPV16 E6 expression	[47]
HCV	RNA	Induced E6AP to increase degradation of HCV viral protein	[41,42]
FeLV	RNA	-	[48]

## Data Availability

Not applicable.

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
