# Peer review of "A Novel Cognition of Decitabine: Insights into Immunomodulation and Antiviral Effects"

_molecules, 2022, doi:10.3390/molecules27061973_

Round 1
Reviewer 1 Report
The manuscript Xiao et. al explains the effects of decitabine on viral diseases and its immunomodulatory and antiviral effects in a short review. This manuscript has merits; however, the way of presentation is inappropriate for the readers. In addition, the manuscript lags some vital information. A major revision is necessary and requests the authors to go through the more detailed comments below.
Line 22-25 needs grammatical corrections (‘considers’ needs to be changed, ‘controlling’ to be changed to ‘regulating’, ‘can help to’ to be changed to ‘could help to’ etc)
Line 83-85 needs correction. The meaning is not clear and check whether the sentence is apt in the context
Line 70: DNA Methyltransferase (DNMT) to be added at the first sentence it is mentioned.
Line 90: IFN-β
Line 58-79: The sentences need to be shortened as those do not come under the heading (something on immune responses). A short description in a small paragraph to be only included. About DNA methylation can be added separately in a paragraph with the heading.
Line 57: The heading needs to modify according to the previous comments.
Line 58 -125: The authors need to very clearly mention with subheading DNA methylation on viral sensors (RIG/MDA-5, Mavs, etc.), TLRs, IRFs, and antiviral responses (genes involved like OAS-1, OAS-2). Also, clearly conclude with the most significant genes identified in the viral diseases as conclusion. Also, find out any previous studies explaining DNA methylation contributes to viral replication stages. It gives the information to the audience in a clear manner.
Line 60: check the sentence is correct and apt to the context
Line 117- 5-AzaC? The abbreviation should be expanded in the first sentence it is mentioned.
Line 122-125: This sentence needs correction.
Line 126: The heading to changed to Decitabine and its role in the immune modulation of viral diseases
Line 159: SK-N-AS NB cells. What is it should be mentioned
Line 126-181: The section needs subheadings like DAC in NF-kB regulation, MAPK signaling, antiviral responses, etc.
Section 4 is well written, though, DAC has been evaluated in some other viruses (eg: dengue, chikungunya, etc.). These must be incorporated and table 1 needs modifications accordingly.
Line 296: Conclusion and Future prospects
Line 310: the first sentence needs to be removed. The starting sentence needs modifications too.
The title of the paper may be changed to ‘Decitabine: Insights into its antiviral and immunomodulatory effects'
Author Response
Response to Reviewer 1 Comments
Dear reviewer:
Thanks for reviewer’s valuable comments and advices on our manuscript. Thank you for your kind letter. We have completely revised the manuscript in accordance with the comments, and carefully proofread the manuscript to minimize typographical, grammatical, and bibliographical errors. In response to the specific questions raised by review, I will reply one by one below.
Point 1: Line 22-25 needs grammatical corrections (‘considers’ needs to be changed, ‘controlling’ to be changed to ‘regulating’, ‘can help to’ to be changed to ‘could help to’ etc)
Response 1: Thank you for your careful comments. Those have been corrected in the latest manuscript.
Point 2: Line 83-85 needs correction. The meaning is not clear and check whether the sentence is apt in the context.
Response 2: We deeply appreciate for your kind advice. Here we are trying to introduce the innate immune response-antiviral innate immune response-regulation of innate immune response step by step. Lines 83-85 may be too descriptive of the innate immune response. We have made deletions and added the description of antiviral innate immune response to further lead to the exploration of DNA methylation on innate immune regulation. We have adjusted accordingly in the latest manuscript.
Point 3: Line 70: DNA Methyltransferase (DNMT) to be added at the first sentence it is mentioned.
Response 3: Thank you for your careful comments. We have put the abbreviation (DNMTs)in line 65 - the first mention of the methyltransferase position.
Point 4: Line 90: IFN-β
Response 4: Thank you for your kind comments. We have made the appropriate changes as suggested.
Point 5: Line 58-79: The sentences need to be shortened as those do not come under the heading (something on immune responses). A short description in a small paragraph to be only included. About DNA methylation can be added separately in a paragraph with the heading.
Response 5: Thank you for your kind advice. We have partially trimmed the section describing the immune response. This paragraph is intended to present the antiviral innate immune response and to elucidate the regulatory role of methylation in the antiviral innate immune response, so it is reasonable to feel that these elements are described together.
Point 6: Line 57: The heading needs to modify according to the previous comments.
Response 6: Thank you for your kind advice. Based on your suggestion, we have made adjustment
Point 7: Line 58 -125: The authors need to very clearly mention with subheading DNA methylation on viral sensors (RIG/MDA-5, Mavs, etc.), TLRs, IRFs, and antiviral responses (genes involved like OAS-1, OAS-2). Also, clearly conclude with the most significant genes identified in the viral diseases as conclusion. Also, find out any previous studies explaining DNA methylation contributes to viral replication stages. It gives the information to the audience in a clear manner.
Response 7: Thank you very much for your suggestions. As you mentioned, the role of DAC on RNA/DNA sensors (RIG/MDA-5, Mavs, etc.), TLRs, IRFs, and antiviral responses (genes involved like OAS-1, OAS-2), etc. needs to be more clearly elucidated. We describe the regulation of RIG by DAC in lines 102-107 (Ref. 15), the regulation of IFR7 by DAC in lines 107-110 (Ref. 2), and the regulation of TLR2/TLR9 by DAC in lines 110-118 (Refs. 21-22). On the one hand, not all of these literature reports investigate the role of these key proteins in antiviral diseases models, while other important proteins related to the regulation of antiviral innate immunity are mainly focused on reports of relevance in proteomic live genomics and do not comprehensively and clearly validate the regulatory mechanisms involved. To address this issue, we also hope that this review will attract more researchers to explore the molecular regulatory mechanism of DAC regulation of antiviral innate immunity. Of course, we have done a lot of research work in this direction (to be published).
Point 8: Line 60: check the sentence is correct and apt to the context
Response 8: Thank you for your kind advice. Echoing the fifth comment, we do think there is some redundancy here, and we have revised it accordingly.
Point 9: Line 117- 5-AzaC? The abbreviation should be expanded in the first sentence it is mentioned.
Response 9: Thank you for your careful comments. We have added the abbreviations here.
Point 10: Line 122-125: This sentence needs correction.
Response 10: Thank you for your kind comments. We have carefully adjusted the description of this sentence.
Point 11: Line 126: The heading to changged to Decitabine and its role in the immune modulation of viral diseases
Response 11: Thank you for your kind comments. Your proposed heading is more appropriate to summarize the content of this paragraph. We have carefully adjusted the description of this sentence.
Point 12: Line 159: SK-N-AS NB cells. What is it should be mentioned
Response 12: Thank you for your kind comments. We have added the abbreviations here.
Point 13: Line 126-181: The section needs subheadings like DAC in NF-kB regulation, MAPK signaling, antiviral responses, etc.
Response 13: Thank you for your kind comments. As an antitumor drug, DAC regulates numerous signaling pathways. In this review, we mainly focus on describing the reported pathways related to DAC in the regulation of innate immune antiviral, and other signaling pathways are briefly mentioned. On one hand, antiviral innate immune response is one of the most important ways to suppress viruses, and on the other hand, antiviral effects of DAC have been reported successively. If each of the DAC-regulated signaling pathways were described as a separate subsection, it might weaken the focus of this review; therefore, we do not think that this section will be divided into multiple subsections to describe them one by one.
Point 14: Section 4 is well written, though, DAC has been evaluated in some other viruses (eg: dengue, chikungunya, etc.). These must be incorporated and table 1 needs modifications accordingly.
Response 14: Thank you for your kind comments. We conducted multiple keyword searches in PubMed, Web of Science, and other websites, and the retrieved articles with direct correlations were mainly those in the table with viruses directly related to DAC, while other viruses may be more descriptive of DAC's similarity to other antiviral drug (nucleic acid analogs) with inhibitory effects on a variety of viruses, but no direct correlations were reported between them.
Point 15: Line 296: Conclusion and Future prospects
Response 15: Thank you for your kind comments. We have made adjustments based on your suggestion.
Point 16: Line 310: the first sentence needs to be removed. The starting sentence needs modifications too.
Response 16: Thank you for your kind comments. We have made adjustments based on your suggestions.
Point 17: The title of the paper may be changed to ‘Decitabine: Insights into its antiviral and immunomodulatory effects'
Response 17: Thank you for your kind advice. We believe that both titles present a similar meaning of full-text summary and retain the original title for now.
Finally, we would like to thank the reviewer again for your detailed and helpful comments on our manuscript. It was a great help to promote our manuscript and of course we hope that our answers will solve your doubts.
Best regards,
Yifei Wang
At 2022-3-14

Reviewer 2 Report
The manuscript entitled "A Novel Cognition of Decitabine: Insights into Immunomodu- 2
lation and Antiviral Effects" is very well presented and deals with topics of great interest.
I recommend the manuscript, as submitted, for publication.
Author Response
Response to Reviewer Comments
Dear reviewer:
We are very grateful to the reviewer for your approval of the content and language organization of our submitted manuscript, and we hope that this review will attract more researchers to the field of decitabine regulation of antiviral innate immune response.
Best regards,
Yifei Wang
At 2022-3-14
Reviewer 3 Report
This is a manuscript discussion about decitabine and its antiviral effects. There are several suggestions for authors to address.
- Authors discussed decitabine as immunomodulator in controlling innate immune response and also anti-HIV activity. However, it's still lack of the connection between innate immune response and antiviral effects.
- Table 1 presented the potential antiviral effects mechanism of decitabine. It made readers confused. Decitabine regulated HIV DNA methylation in HIV infected cells, but decitabine unregulated host cell (human liver cell) ApoA1 in HBV infected population. Authors might separate into 2 tables.
Author Response
Response to Reviewer Comments
Dear reviewer:
Thank you for your kind letter. In response to the specific questions raised by review, we will reply one by one below.
Point 1: Authors discussed decitabine as immunomodulator in controlling innate immune response and also anti-HIV activity. However, it's still lack of the connection between innate immune response and antiviral effects.
Response 1: Thank you very much for your valuable comments. We couldn't agree with you more. Indeed, the precise relationship and role of decitabine in the regulation of innate immunity and antiviral response is still poorly understood. In this manuscript, we review the reported antiviral effects of decitabine and the studies related to the regulation of innate immune responses by decitabine. As for the clear function and role of decitabine in modulating antiviral innate immune response, further experimental validation is needed. And in this regard, we have some important experimental results to be published soon. This will help us to further understand the molecular mechanisms involved. This manuscript also aims to attract more researchers' attention to the study of decitabine in modulating antiviral innate immune response and to propose more ideas and theoretical support for the development of antiviral drugs.
Point 2: Table 1 presented the potential antiviral effects mechanism of decitabine. It made readers confused. Decitabine regulated HIV DNA methylation in HIV infected cells, but decitabine unregulated host cell (human liver cell) ApoA1 in HBV infected population. Authors might separate into 2 tables.
Response 2: Thank you for your kind advice. As a key protein in the host regulation of viral infection, ApoA1 is regulated by both HBV viruses and host proteins DNMTs, following the tendency to be a dynamic effect (manuscript lines 357-365, reference 40). Decitabine acts as an inhibitor of DMNTs, and we believe that decitabine acts more by inhibiting DNMTs and promoting ApoA1 expression (in contrast to viral protein function). It is considered more appropriate to put in the same table to describe the relationship of decitabine regulation of viruses.
Thanks again to the reviewer for your comments on this manuscript.
Best regards,
Yifei Wang
At 2022-3-14

Round 2
Reviewer 1 Report
The authors addressed most of the queries pretty well and deserve acceptance. Congratulations.